# Assessment of Phenolic Acid Content and Antioxidant Properties of the Pulp of Five Pumpkin Species Cultivated in Southeastern Poland

**DOI:** 10.3390/ijms24108621

**Published:** 2023-05-11

**Authors:** Małgorzata Stryjecka, Barbara Krochmal-Marczak, Tomasz Cebulak, Anna Kiełtyka-Dadasiewicz

**Affiliations:** 1Institute of Human Nutrition and Agriculture, The University College of Applied Sciences in Chełm, 22-100 Chełm, Poland; mstryjecka@panschelm.edu.pl; 2Department of Plant Production and Food Safety, The University College of Applied Sciences in Krosno, 38-400 Krosno, Poland; 3Department of Food Technology and Human Nutrition, Institute of Food Technology and Nutrition, College of Natural Sciences, University of Rzeszów, 35-601 Rzeszów, Poland; 4Department of Plant Production Technology and Commodity Sciences, University of Life Sciences, 20-950 Lublin, Poland

**Keywords:** *Cucurbita*, species, β-carotene, L-ascorbic acid, vitamin E, phenolic compounds, antioxidant activities

## Abstract

Antioxidant properties and phenolic acid content in the pulp of five pumpkin species were evaluated. The following species cultivated in Poland were included: *Cucurbita maxima* ‘Bambino’, *Cucurbita pepo* ‘Kamo Kamo’, *Cucurbita moschata* ‘Butternut’, *Cucurbita ficifolia* ‘Chilacayote Squash’, and *Cucurbita argyrosperma* ‘Chinese Alphabet’. The content of polyphenolic compounds was determined by ultra-high performance liquid chromatography coupled with HPLC, while the total content of phenols and flavonoids and antioxidant properties were determined by spectrophotometric methods. Ten phenolic compounds (protocatechuic acid, p-hydroxybenzoic acid, catechin, chlorogenic acid, caffeic acid, p-coumaric acid, syringic acid, ferulic acid, salicylic acid, kaempferol) were identified. Phenolic acids were the most abundant compounds; the amount of syringic acid was found to be the highest, ranging from 0.44 (*C. ficifolia*) to 6.61 mg∙100 g^−1^ FW (*C. moschata*). Moreover, two flavonoids were detected: catechin and kaempferol. They were found at their highest level of content in *C. moschata* pulp (catechins: 0.31 mg∙100 g^−1^ FW; kaempferol: 0.06 mg∙100 g^−1^ FW), with the lowest amount detected in *C. ficifolia* (catechins: 0.15 mg∙100 g^−1^ FW; kaempferol below the limit of detection). Analysis of antioxidant potential showed significant differences depending on the species and the test used. The DPPH radical scavenging activity of *C. maxima* was 1.03 times higher than *C. ficiofilia* pulp and 11.60 times higher than *C. pepo*. In the case of the FRAP assay, the multiplicity of FRAP radical activity in *C. maxima* pulp was 4.65 times higher than *C. Pepo* pulp and only 1.08 times higher compared to *C. ficifolia* pulp. The study findings show the high health-promoting value of pumpkin pulp; however, the content of phenolic acids and antioxidant properties are species dependent.

## 1. Introduction

Antioxidant properties found in plant products have beneficial effects on human health. Their role is to prevent the harmful effects of free radicals, which at high concentrations pose a serious threat to cell structures [1]. Many studies have demonstrated that a diet rich in plant-based products plays an important role in the prevention of many diseases, such as diabetes mellitus, atherosclerosis, Alzheimer’s disease, Parkinson’s disease, and neoplastic diseases [2,3,4,5,6]. The health-promoting properties of some plant raw materials result from a high amount of antioxidant compounds, vitamins C, E, and A, carotenoids, organic acids, and selenium [7,8]. One of the plant-based raw materials characterized by a high amount of bioactive compounds is pumpkin pulp [9,10,11].

Pumpkin (*Cucurbita* L.) is an annual plant belonging to the *Cucurbitaceae* family, which includes about 130 genera and over 800 species [12]. Pumpkin is a vegetable that is easy to grow, and its yields are high. Currently, its cultivation is widespread throughout the world [11,13]. In many countries, pumpkin is used as a medicine for its anti-inflammatory, antioxidant, antiviral, and antidiabetic properties. It is particularly popular in Austria, Hungary, Mexico, Slovenia, China, Spain, and various European, Asian, and African countries [14,15]. The raw material of pumpkin is peel, pulp, and seeds. Raw pumpkin pulp of pumpkin fruits is used to produce many preserves, marinades, jams, sauces, and cakes, and is also a component of various food products for adults and children [15]. Pumpkin varieties with orange pulp, indicating they contain high amounts of carotene (mainly α and β), are most valuable for the processing industry. Carotene content in the fresh weight of pumpkin fruits varies from 2 to 10 mg∙100 g^−1^, though in some varieties it may exceed 22 mg∙100 g^−1^ [3]. Carotenoids found in pumpkin pulp protect tissues from the damage caused by free radicals and, consequently, from the development of cancer, premature aging, cataracts, age-related macular degeneration, atherosclerosis, and many other degenerative diseases [16]. Pumpkin pulp also contains small amounts of vitamins C (from 9 to 10 mg∙100 g^−1^), E (from 1.03 to 1.06 mg∙100 g^−1^), and B6 (from 0.06 to 0.11 mg∙100 g^−1^), as well as thiamine (0.05 mg∙100 g^−1^), riboflavin (0.11 mg∙100 g^−1^), niacin (0.6 mg∙100 g^−1^), K (1.1 μg∙100 g^−1^), and folate (from 0.16 to 0.20 μg∙100 g^−1^) [10,11,12]. Moreover, pumpkin pulp is a valuable source of minerals such as potassium, phosphorus, magnesium, iron, and selenium [17]. It may also have antidiabetic, antihypertensive, anticancer, immunomodulatory, anti-inflammatory, antimicrobial, and antiparasitic effects [18]. The bioactive substances found in pumpkin pulp show diuretic and detoxifying effects, regulate metabolism, and are used to treat kidney and cardiovascular diseases [19,20]. A study by Yadav et al. [19] confirms that pumpkin pulp has significant medicinal properties that can be used to treat diabetes and also reduce its incidence. Other potential medical benefits of raw pumpkin pulp, such as hepatoprotective and anticancer effects, have also been reported in scientific reports by these authors [19]. Pumpkin extracts protect mucosal cells from free radicals and reduce the effects of indomethacin and the occurrence of inflammation and ulceration in both the stomach and small intestine [21]. Extracts of raw pumpkin flesh show strong antimicrobial potential against three bacterial species (*E. coli*, *S aureus*, and *P. multocida*) [22]. Pumpkin pulp is also used to produce a natural pigment powder that is added to confectionery, baked goods, pasta, and dairy products [23]. According to a study by Amin et al. [24], pumpkin pulp is a good source of nutrients and can be considered a nutraceutical. Diets based on pumpkin provide rich sources of carotenoids and can prevent skin diseases, eye diseases, and cancer. The inclusion of pumpkin as an ingredient and source of β-carotene in various foods is considered very effective for vitamin A-related health disorders [25]. Numerous available scientific studies confirm that the nutrient content of pumpkin flesh varies genetically. In addition, the composition and level of bioactive compounds varies depending on their origin and climatic and soil conditions [19,20,26]. Considering the fact that more and more new varieties and species of pumpkin that differ in shape, color, and taste are being cultivated and sold, we undertook a study to gain knowledge about antioxidant properties and the levels of phenolic acid in the fresh flesh of five pumpkin species grown in southeastern Poland. An additional goal of our research is to provide knowledge of raw pumpkin flesh as a valuable raw material for the pharmaceutical, cosmetic, and food industries.

## 2. Results and Discussion

### 2.1. Content of β-Carotene

The content of β-carotene in the samples is shown in Table 1. The content of β-carotene in pumpkin pulp extracts varied according to individual pumpkin species and ranged from 0.21 (12.4%) (*C. pepo*) to 1.73 (100%) (*C. maxima*) mg∙100 g^−1^ FW. In the study by Zhou et al. [27], the content of β-carotene in three pumpkin species (*C. maxima*, *C. pepo*, and *C. moschata*) grown in China was slightly lower than that in our study, ranging from 0.13 to 1.67 mg∙100 g^−1^ FW. In the pumpkin species tested in our study but cultivated in Austria, the content of β-carotene ranged from 0.06 to 7.4 mg∙100 g^−1^ FW [16]. According to Husain et al. [28], the content of β-carotene in *C. maxima* pulp was 6.18 mg∙100 g^−1^.

### 2.2. Content of L-Ascorbic Acid and Vitamin E

The content of L-ascorbic acid and vitamin E in the pulp of the pumpkin species tested is presented in Table 1. The content of L-ascorbic acid ranged from 6.4 (*C. moschata*) to 14.7 (*C. maxima*) mg∙100 g^−1^ FW. Likewise, the highest content of vitamin E was found in *C. maxima* (1.59 mg∙100 g^−1^ FW), while the lowest content was determined in *C. ficifolia* (0.38 mg∙100 g^−1^ FW). Similar results were reported by Zhou et al. [27]. In the study by Takizawa et al. [29] assessing *Cucurbita moschata*, the content of ascorbic acid was 6.8 mg∙100 g^−1^ FW. According to Roura et al. [30], the content of L-ascorbic acid in mature *C. moschata* Duch. fruits cultivated in Argentina was 22.87 mg∙100 g^−1^ FW. The L-ascorbic acid content reported by them is significantly higher compared to our findings, which can be attributed to differences in the climate–soil conditions of pumpkin cultivation.

### 2.3. Flavonoid Content

The content of flavonoids in the pulp of the pumpkin species assessed ranged from 0.75 (*C. pepo*) to 8.73 (*C. maxima)* mg QE∙100 g^−1^ FW (Table 1). Similar results for *C. maxima* (8.23 mg QE∙100 g^−1^ FW), *C. pepo* (0.51 mg QE∙100 g^−1^ FW), and *C. moschata* (5.36 mg QE 100 g^−1^ FW) were reported by Zhou et al. [27]. According to Rakass et al. [31], the content of flavonoids in the pulp of *C. maxima* was 2.21 mg CE∙g^−1^ extract. Furthermore, in the study by Hussain et al. [28], the total content of flavonoids in the pulp of *C. maxima* was 139.37 CE∙100 g^−1^ powder. 

### 2.4. Total Polyphenol Content 

The highest total polyphenol content was found in *Cucurbita moschata* (477.89 mg GAE∙100 g^−1^ FW), while the lowest was found in *Cucurbita pepo* (62.91 mg GAE∙100 g^−1^ FW) (Table 1). Our findings were similar to those reported by Zhau et al. [27]; significantly higher results were presented by Azizah et al. [32] (90 mg GAE∙100 g^−1^ FW), Nawirska-Olszańska et al. [33] (24 mg GAE∙100 g^−1^ FW), and Oloyede et al. [34] (23.7 mg 100 g^−1^). According to Yadav et al. [19], *Cucurbita pepo* was characterized by several times higher total polyphenol content compared to *Cucurbita maxima*. Moreover, in the study by Telesiński et al. [35] analyzing the total polyphenol content in four varieties of *Cucurbita moschata*, the highest polyphenol content was demonstrated in ‘Kurinishiki’ (99.86 mg GAE∙100 g^−1^ FW), while lower content was found in ‘Muskatna’ (80.41 mg GAE∙100 g^−1^ FW), ‘Butternut Rugosa’ (79.23 mg GAE∙100 g^−1^ FW), and ‘Muscade de Provence’ (73.70 mg GAE∙100 g^−1^ FW). 

### 2.5. Content of Phenolic Compounds

Ten phenolic compounds were identified in the study samples of the pumpkin pulp, i.e., protocatechuic acid, p-hydroxybenzoic acid, catechin, chlorogenic acid, caffeic acid, p-coumaric acid, syringic acid, ferulic acid, salicylic acid, and kaempferol (Table 2). Phenolic acids were the most abundant compounds; the amount of syringic acid was found to be the highest, ranging from 0.44 (*C. ficifolia*) to 6.61 mg∙100 g^−1^ FW (*C. moschata*). The lowest values were noted for chlorogenic, caffeic, p-coumaric, and ferulic acid, whose levels in *C. ficifolia* and *C. argyrosperma* were below detection limits. The next most abundant phenolic acid was protocatechuic acid, whose content varied in individual pumpkin species, ranging from 0.34 mg∙100 g^−1^ FW (*C. ficifolia*) to 1.37 mg∙100 g^−1^ FW (*C. moschata*). Comparable results were reported by Kostecka-Gugała et al. [36]. Pumpkin is a richer source of ferulic acid compared to other vegetables and fruits. Ferulic acid is known to have anti-neoplastic properties due to its effects on the vascular endothelial growth factor (VEGF), platelet-derived growth factor (PDGF), and hypoxia-inducible factor-1 (HIF-1) [37]. Moreover, ferulic acid can inhibit skin photoaging; therefore, it is used as a component of cosmetic preparations. Pumpkin pulp was also found to contain salicylic acid, and though its antioxidant properties are moderate, it can reduce the risk of myocardial infarction and ischemic stroke. Additionally, although the content of ferulic acid in pumpkin fruits is significantly lower than, for example, raspberries, pumpkin pulp can provide continuous supplementation with this compound due to the long storage time and the significantly higher reserves available in the food processing industry [38].

Moreover, two flavonoids, catechin and kaempferol, were detected in the samples of various pumpkin species. The highest content of these compounds was found in *C. moschata* (catechin: 0.31 mg∙100 g^−1^ FW; kaempferol: 0.06 mg∙100 g^−1^ FW), while the lowest amount was determined in *C. ficifolia* (catechin: 0.15 mg∙100 g^−1^ FW; kaempferol below detection limits). Flavonoids are potent antioxidants characterized by defined health properties: they can protect against oxidative stress-induced diseases, modulate the activity of enzymes, and interact with some receptors [39]. Moreover, flavonoids protect against cardiovascular diseases by reducing the rate of oxidation of low-density lipoproteins [34]. Currently, scientists are paying a great deal of attention to kaempferol, which is considered a potential agent to treat cancer due to its strong capacity to reduce oxidative stress [40,41].

### 2.6. Antioxidant Properties

DPPH free radical scavenging activity in the pulp of the pumpkin species tested proved to be statistically significant at *p* = 0.05 (Table 1). Statistically significant differences were observed among all studied pumpkin species. The extent of changes ranged from 268 to 3117 TEAC mmol L^−1^ Trolox. DPPH radical scavenging activity for *C. maxima* was 1.03 times higher than *C. ficiofilia* and 11.6 times higher than *C. pepo*. In the case of the FRAP assay, the multiplicity of FRAP radical activity in the pulp *of C. maxima* was 4.65 times higher than in *C. pepo*, yet only 1.08 times higher than in *C. ficifolia*. The highest amount of content in terms of study parameters (ascorbic acid, vitamin E, β-carotene, polyphenols, total flavonoids, DPPH radicals, and FRAP radicals) was observed in *C. maxima*. Jiao et al. [42] also demonstrated that free radical scavenging activity is strictly associated with the concentration of β-carotene, and our results are similar. Analysis of the antioxidant potential of pumpkin pulp showed significant variation depending on the species and the test applied (Table 1, Figure 1 and Figure 2). A high amount of phenolic compounds does not always indicate high total antioxidant potential. The antioxidant properties of pumpkin pulp can also be affected by the presence of carotenoids and vitamins, especially vitamins C and E [11]. Therefore, it is extremely difficult to estimate antioxidant potential based exclusively on the content of individual bioactive compounds.

In a study by Kulczyński et al. [43], the highest antioxidative potential in aqueous extracts was observed in ‘Miranda’ belonging to *C. pepo* (105.96 mg Troloxu∙100 g^−1^), while the lowest potential was found in ‘Festival’ (25.54 mg Trolox∙100 g^−1^) and ‘Snow Ball’ (24.68 mg Troloxu∙100 g^−1^), also belonging to *C. pepo*. Telesiński et al. [35] analyzed the antioxidant properties of the pulp of four pumpkin varieties belonging to *Cucurbita moschata:* ‘Kurinishiki’, ‘Butternut Rugosa’, ‘Muscade de Provence’, and ‘Muscatna’. The highest DPPH radical scavenging ability was demonstrated in ‘Kurinishiki’ (31.39% inhibition). The lowest antioxidative potential was found in ‘Muscatna’ (17.41%). According to Oyeleke et al. [44], the pulp of *Cucurbita mixta* is characterized by higher DPPH radical scavenging ability compared to the pulp of *Cucurbita maxima*.

### 2.7. Cluster Analysis and Principal Component Analysis (PCA) as a Graphical Interpretation of Results

Statistical analysis carried out to determine correlations among the quality characteristics of pumpkin fruit pulp (Figure 1a,b and Figure 2a,b) allowed us to demonstrate the hidden data structure in the form of clusters (cluster analysis and PCA). The reduction of qualitative data concerning antioxidative activity (ascorbic acid, vitamin E, β-carotene, total polyphenols, flavonoids, DPPH, FRAP) allowed us to distinguish two groups of pumpkin pulp. As shown in Figure 1a,b, the first cluster of species with similar quality characteristics included two species: *C. pepo* and *C. argyrosperma*. The second cluster was formed by variables characterizing the level of the antioxidative activity of the pulp of *C. ficifandlia, C. moschata*, and *C. maxima*, and the pulp of the above species was thus characterized by higher antioxidative activity. A closer look at the clusters in the dendrogram shows that the first cluster contained species whose pulp showed higher antioxidative activity when compared to the pulp in the second cluster. Dendrogram 2a and 2b show clusters of the content of phenolic compounds in the pulp of the pumpkin species tested. The reduction of data revealed their new structure, which amounts to the separation of three clusters showing interdependencies (correlations). 

However, in this case, the varieties with the highest amount of phenolic compounds in the pulp were found to be *C. pepo* and *C. moschata*, followed by *C maxima*. The lowest amount of phenolic compounds was determined in *C. ficifolia* and *C. argyrosperma*.

## 3. Materials and Methods

### 3.1. Plant Material

The study material was the pulp of five pumpkin species: *Cucurbita maxima* “Bambino”, *Cucurbita pepo* “Kamo Kamo”, *Cucurbita moschata* “Butternut”, *Cucurbita ficifolia* “Chilacayote Squash”, and *Cucurbita argyrosperma* “Chinese Alphabet”.

Plants were grown in Żyznów (49°49′ N 21°50′ E) (Poland) in 2018 under identical agro-climatic conditions.

The pumpkin fruits were harvested on October 20 by cutting off the fruits from the shoot. Immediately after harvesting, the pumpkin fruits were washed and peeled, and seeds were then removed. The resulting pulp was crushed into small pieces, which were frozen at −25 °C and then freeze-dried (0.37 mBa) for 48 h in a FreeZone 12 L freeze dryer (Labconco Corporation, Kansas City, MO, USA). The obtained samples of lyophilizates were stored at −25 °C in dark jars protected from light [36].

### 3.2. Sample Preparation 

The extracts were prepared by crushing 0.75 g of freeze-dried material in a mortar with 80% methanol (Sigma-Aldrich, St. Louis, MO, USA) poured in portions to a total volume of 25 mL. The extraction time was 4 h. Each sample was filtered through a funnel with a sintered disc into a 25 mL volumetric flask. The extraction process was carried out at a temperature of 22 °C with limited lighting. Three extracts of pulp were prepared for each pumpkin species. The extracts were stored in the dark at −25 °C for one week [36]. 

### 3.3. Determination of β-Carotene Content 

The content of β-carotene was determined using an HPLC technique based on the method of Liu et al. [20] with minor modifications. A volume of 30 mL of acetone was added to 15 g of the sample and was homogenized for 15 min using a UP200St_TD ultrasonic homogenizer (Hielscher Ultrasonics, Teltow, Germany). Subsequently, the resulting sample was centrifuged at 10,000 rpm at 4 °C for 10 min. The extracts obtained were collected and supplemented with acetone to a volume of 100 mL. A volume of 50 mL of KOH (10%) was then dissolved in methanol and added to the extracts for saponification in a water bath at 45 °C for 50 min. After this time, all extracts were added to 100 mL of petroleum ether and mixed; the organic layer formed was dehydrated, concentrated using a column with anhydrous sodium sulphate, and evaporated to dryness. The dehydrated residue was dissolved in hexane and filtered through a membrane filter (0.45 μm) for analysis with the RF-10AXL HPLC system; measurements were carried out at 445 nm at 30 °C (Shimadzu, Kyoto, Japan). The Waters Sunfire TM C18 analytical column (4.6 × 9 × 250 mm i.d., 5 μm particle size) was used. The mobile phase was acetonitrile:methanol:methylene chloride (6:2:2, *v*/*v*/*v*), with isocratic flow at a rate of 1.0 mL∙min^−1^. The concentration of β-carotene in the samples was calculated based on the external standard of β-carotene and is presented as a milligram of β-carotene per 100 g^−1^ FW.

### 3.4. Determination of L-Ascorbic Acid Content

The content of L-ascorbic acid was determined by titration with 2,6-dichlorophenolindophenol according to the method described by Zhou et al. [27] with minor modifications. A volume of 50 mL of 0.02 g∙mL^−1^ oxalic acid solution was added to 50 g of the sample and homogenization was carried out (IKA Ultra Turrax homogeniser, Warsaw, Poland). Subsequently, the sample was centrifuged at 4000 rpm at 4 °C for 15 min. The obtained supernatant (10 mL) was titrated with a solution of 2,6-dichlorophenolindophenol (concentration—0.1%). The concentration of L-ascorbic acid was calculated from the volume of 2,6-dichlorophenolindophenol used for titration and expressed in milligrams per 100 g fresh weight.

### 3.5. Determination of Total Phenols

The content of phenolic compounds in the extracts was determined based on the reaction with the Folin–Ciocalteu reagent [45]. The extract (0.25 mL) was mixed with 0.25 mL of 25% Na_2_CO_3_, 125 mL of the Folin–Ciocalteu reagent (Sigma-Aldrich, twice diluted with distilled water before analysis), and 2.25 mL of water and then incubated for 20 min. Absorbance was measured spectrophotometrically at a wavelength of 765 nm (UV-2600i, Shimadzu, Japan). The total phenolic content of pumpkin pulp extracts was expressed as mg of gallic acid equivalents (GAE) per 100 g fresh weight.

### 3.6. Total Flavonoid Content

Total flavonoid content was determined using the method described by Di Marco et al. [46] with minor modifications. A volume of 1 mL of deionized water was added to 0.1 g of the sample tested. Subsequently, 0.5 mL of the resulting sample was dissolved in 95% ethyl alcohol. The next step was to add 1 mL of 10% aluminum chloride hexahydrate, 0.1 mL of 1 M potassium acetate, and 2.8 mL of deionized water to the sample. The resulting sample was incubated at 25 °C for 50 min. The absorbance of the reaction mixture was determined at a wavelength of 415 nm using a spectrophotometer (UV-2600i, Shimadzu, Japan). Deionized water and quercetin were set as a blank and standard, respectively. Using a seven-point standard curve (0–50 mg∙L^−1^), total flavonoid content was determined in all samples in triplicate. The results were expressed in milligrams of quercetin equivalents (QE) per 100 g fresh weight.

### 3.7. Identification of Phenolic Compounds by HPLC 

To identify phenolic compounds in pumpkin pulp extracts, the high-performance liquid chromatography (HPLC) method (Shimadzu LC-10AS chromatograph equipped with a C18 RP column and an SPD-10AV UV-VIS detector) was used. Signal detection was set at wavelengths of 325 and 265 nm. Chromatographic separation was carried out at 33°C using the following solvents: (A) water (Sigma-Aldrich) with acetic acid (0.1%), (B) methanol (Sigma-Aldrich, ultra-pure) with acetic acid (0.1%). The following gradients were used: 90% A, 10% B for 20 min; 75% A, 25% B for 30 min; 65% A, 35% B for 40 min; 55% A, 54% B for 50 min; 50% A, 50% B for 60 min; 30% A, 70% B for 62 min; 100% B up to 80 min; and 80% A, 10% B up to 85 min. The flow rate was 1 mL min^−1^. The identification of phenols was based on the retention times of chlorogenic, caffeic, p-hydroxybenzoic, p-coumaric, ferulic, protocatechuic, syringic (Sigma-Aldrich), and salicylic (LGC Standards) acid, as well as retention of (+)-catechin (LGC Standards) and kaempferol (Sigma-Aldrich) [36].

### 3.8. Determination of Antioxidant Properties by DPPH Assay

The antioxidant capacity of extracts was assessed based on the reduction of the synthetic and stable 2,2-diphenyl-1-picrylhydrazyl (DPPH^●^) free radical. The colorimetric method allows for the measurement of changes in the absorbance of a DPPH solution at a wavelength of 517 nm as the result of the antioxidant activity of the sample [47]. A volume of 2.8 mL of 0.1 mM DPPH solution (Sigma-Aldrich, St. Louis, MO, USA) in 96% ethanol was mixed with 0.2 mL of the pumpkin extract. Absorbance was measured after 30 min of incubation using a spectrophotometer (UV-2600i, Shimadzu, Japan). The results were expressed as μmol Trolox (Sigma-Aldrich, St. Louis, MO, USA) per 100 g fresh weight (TEAC μ mol∙^L−1^ Trolox FW).

### 3.9. Determination of Antioxidant Properties by FRAP Assay

A fresh FRAP solution consisting of 25 mL of 0.3 mol∙L^−1^ acetate buffer (pH 3.6), 2.5 mL of 10 mmol∙L^−1^ of TPTZ (soluble in 40 mmol∙L^−1^ HCl), and 2.5 mL of 20 mmol∙ L^−1^ ferric chloride was prepared [48]. The sample was centrifuged at 10,000 rpm^−1^ at 4 °C for 20 min. The collected supernatants were mixed. For further determination, 0.4 mL of ten-fold-diluted supernatant mixed with 8.0 mL of FRAP solution (0.14 mmol∙L^−1^) was used, which was mixed and left at 37 °C for 15 min. Absorbance was measured at 593 nm using a spectrophotometer (UV-2600i, Shimadzu, Japan); the FRAP solution was a blank. The results were expressed as μmol Trolox (Sigma-Aldrich, St Louis, MO, USA) per 100 g fresh weight (TEAC μmol∙L^−1^ Trolox FW).

### 3.10. Determination of Vitamin E

To achieve better extraction of vitamin E, 2 mL of distilled water was added to 2 g of the crumbled sample. After 5 min of soaking, 200 mg of MgCO_3_ and 200 mg of Na _2_SO_4_, followed by 10 mL of methanol/tetrahydrofuran (1 + 1, *v*/*v*; + 0.1% BHT, *v*/*w*) and an internal standard, α-tocopherol acetate (100 μL, 1 mg∙mL^−1^), were added to the samples. The extraction was carried out in an ultrasonic bath (INTERSONIC I-10, Olsztyn, Poland) in water for 15 min and was repeated two times, with 10 mL of the solvent used each time. Subsequently, the samples extracts were centrifuged at 4000 rpm min^−1^ for 10 min. The combined extracts were concentrated in a rotary evaporator at 30 ± 1 °C for about 30 min to a volume of about 1 mL and dissolved in 10 mL of n-hexane/methyl-tert-butyl ether (98/2; *v*/*w*). Vitamin E (tocopherols and tocotrienols) was determined according to the method described by Franke et al. [49] using an NP-HPLC system (Merck Hitachi KgaA, Darmstadt, Germany). The system consisted of an L-6200 pump, an AS-2000 autosampler, an F-1080 fluorescence detector, and a D-6000 interface. Separation was carried out on a Eurospher 100–5 Diol 5 μm Vertex column (250 × 4.00 mm) (KNAUER GmbH, Berlin, Germany). The column was heated to 35 ± 1 °C. Isocratic elution was carried out with n-hexane/methyl-tert-butyl ether (98/2; *v*/*w*). The volume of injections was 20 μL at a flow rate of 1.5 mL∙min^−1^ and a total time of 45 min. The measurement was performed at an excitation wavelength of 292 nm and an emission wavelength of 330 nm using a fluorescence detector. To identify the substances, their retention times were compared with those of external standards. The content of individual vitamin E derivatives is presented in mg∙100 g ^−1^ fresh weight, and the content of vitamin E is presented as the sum of all derivatives.

### 3.11. Statistical Analysis

The results are presented as the mean of three independent measurements. All analyses were performed in Statistica version 13.3 (StatSoft, Krakow, Poland). The significance of means was examined post hoc by the Duncan’s test, which is a part of one-way analysis of variance (ANOVA). *p* < 0.05 was considered significant. In order to establish deeper links between the analyzed variables, a data reduction technique based on cluster analysis and PCA (principal component analysis) was used.

## 4. Conclusions

Our study findings demonstrate that pumpkin pulp is a source of phenolic acids and has antioxidant properties. However, the total content of the individual compounds varied and was species dependent. Ten phenolic compounds were identified in the pulp of each pumpkin species: protocatechuic acid, p-hydroxybenzoic acid, catechin, chlorogenic acid, caffeic acid, p-coumaric acid, syringic acid, ferulic acid, salicylic acid, and kaempferol. Phenolic acids were most abundant, including syringic acid. The lowest values were obtained for chlorogenic, caffeic, p-coumaric, and ferulic acid, whose levels were below the limit of detection in *C. ficifolia* and *C. argyrosperma*. Moreover, two flavonoids, catechin and kaempferol, were detected in the pumpkin pulp; however, their levels were different in the individual pumpkin species. Analysis of the antioxidant potential of pumpkin pulp showed significant variation depending on the species and the test used. The highest antioxidative potential in all tests was found in *C. Maxima*, while the lowest values were demonstrated in the pulp of *C. pepo*. In our study, the content of the compounds analyzed varied in individual pumpkin species. The results of our study are a valuable source of information on the varying levels of antioxidant properties and phenolic acids of the flesh of the studied pumpkin species grown under the soil and climatic conditions of Poland. In addition, the results of our research are of an applied nature, as they provide the pharmaceutical and cosmetic industries with knowledge of the health-promoting properties of pumpkin flesh as a valuable raw material for the production of extracts that can be used in the development of new medicinal products, as well as a valuable source of information for food processing for the development of new food products, including functional foods in both adult and child nutrition. The research results we have presented will increase consumer awareness of the health-promoting properties of various pumpkin species and can also be a source of knowledge regarding the selection for cultivation of pumpkin species with the highest antioxidant properties and the highest phenolic acid content. The published results of our study may also contribute to further research on these pumpkin species for health benefits.

## Figures and Tables

**Figure 1 ijms-24-08621-f001:**
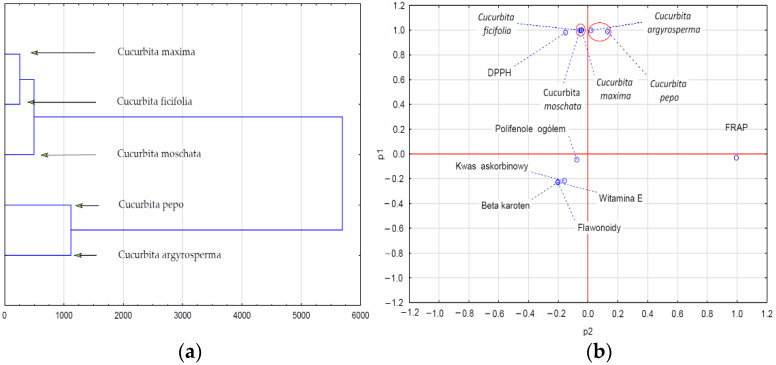
Analysis of clusters (**a**) and PCA (**b**) of the antioxidative activity of the pulp of pumpkin species.

**Figure 2 ijms-24-08621-f002:**
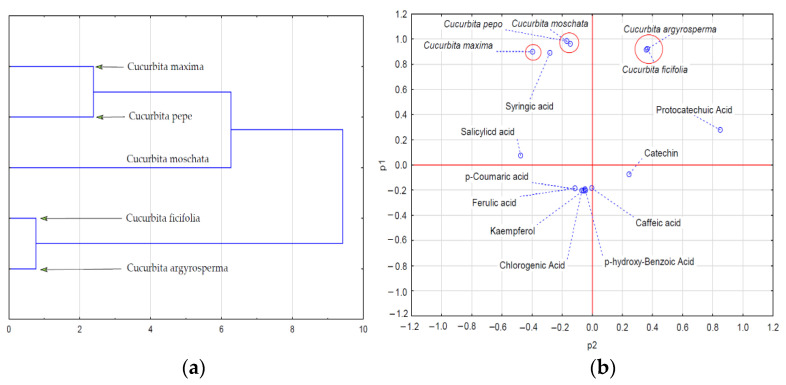
Analysis of clusters (**a**) and PCA (**b**) of the content of phenolic compounds in the pulp of pumpkin species.

**Table 1 ijms-24-08621-t001:** Antioxidant activities and the content of β-carotene, L-ascorbic acid, and total phenols in five species of pumpkin pulp.

Species of *Cucurbita*	L-Ascorbic Acid (mg∙100 g^−1^ FW)	Vitamin E (mg∙100 g^−1^ FW)	Total Phenols (mgGAE∙100 g^−1^ FW)	Total Flavonoids (mg QE∙100 g^−1^ FW)	β-Carotene (mg∙100 g^−1^ FW)	Antioxidant Activities TEAC (μmol∙L^−1^ Trolox)	
						DPPH	FRAP
^a^ *C. maxima* ‘Bambino’	14.7 ± 0.07 ^b,c,d,e^	1.59 ± 0.04 ^b,c,d,e^	443.82 ± 1.5 ^b,c,d,e^	8.73 ± 0.03 ^b,c,d,e^	1.73 ± 0.14 ^b,c,d,e^	3117.84 ± 2.84 ^b,c,d,e^	377.82 ± 1.00 ^b,c,d,e^
^b^ *C. pepo* ‘Kamo Kamo’	8.8 ± 0.07 ^a,c,e^	0.44 ± 0.01 ^a,c,^	62.91 ± 1.2 ^a,c,d,e^	0.75 ± 0.06 ^a,c,d,e^	0.21 ± 0.01 ^a,c^	268.97 ± 1.50 ^a,c,d,e^	81.27 ± 1.00 ^a,c,d,e^
^c^ *C. moschata* ‘Butternut’	6.4 ± 0.09 ^a,b,d,e^	0.72 ± 0.02 ^a,b,d,e^	477.89 ± 2.0 ^a,b,d,e^	6.11 ± 0.01 ^a,b,d,e^	0.65 ± 0.05 ^a,b,d,e^	2778.57 ± 1.32 ^a,b,d,e^	311.28 ± 0.28 ^a,b,d,e^
^d^ *C. ficifolia* ‘Chilacayote Squash’	9.73 ± 0.06 ^a,c,e^	0.38 ± 0.02 ^a,c,e^	339.86 ± 1.24 ^a,b,c,e^	1.39 ± 0.01 ^a,b,c,e^	0.25 ± 0.05 ^a,c^	3009.74 ± 2.00 ^a,b,c,e^	349.88 ± 1.88 ^a,b,c,e^
^e^ *C. argyrosperma* ‘Chinese Alphabet’	11.71 ± 0.09 ^a,b,c,d^	0.33 ± 0.03 ^a,c,d^	117.47 ± 1.26 ^a,b,c,d^	2.17 ± 0.07 ^a,b,c,d^	0.22 ± 0.02 ^a,c^	1192.31 ± 1.31 ^a,b,c,d^	214.57 ± 1.57 ^a,b,c,d^

Statistically significant differences in means are marked with letters. The test of differences among group means was conducted at *p* < 0.05 using the Scheffe’s post hoc procedure testing multiple comparisons available in ANOVA Statistica 13. ±SD—standard deviation.

**Table 2 ijms-24-08621-t002:** Content of selected phenolic compounds in the pulp of five *Cucurbita* species.

Phenolic Compounds (mg∙100 g^−1^ FW)	Species				
	*C. maxima* ^a^ ‘Bambino’	*C. pepo* ^b^ ‘Kamo Kamo’	*C. moschata* ^c^ ‘Butternut’	*C. ficifolia* ^d^ ‘Chilacayote Squash’	*C. argyrosperma* ^e^ ‘Chinese Alphabet’
protocatechuic acid	0.42 ^bc^ ± 0.02	1.01 ^a,b^ ± 0.02	1.37 ^a,d,e^ ± 0.10	0.34 ^b,c,e^ ± 0.03	0.64 ^b,c^ ± 0.05 ^c^
p-hydroxy-benzoic acid	0.008 ^c,e^ ± 0.00	0.002 ^c,e^ ± 0.00	0.01 ^b,d^ ± 0.00	0.003 ^c,e^ ± 0.00	0.01 ^b,d^ ± 0.00
catechin	0.16 ^c^ ± 0.01	0.16 ^c^ ± 0.02	0.31 ^b^ ± 0.04	0.15 ^b^ ± 0.05	0.11 ^b^ ± 0.00
chlorogenic acid	0.01 ^b^ ± 0.00	0.07 ^a,c,d,e^ ± 0.01	0.02 ^b^ ± 0.00	- ^b,^*	- ^b,^*
caffeic acid	0.04 ^b,c,e^ ± 0.00	0.02 ^c^ ± 0.00	0.06 ^b,d,e^ ± 0.00	0.03 ^c^ ± 0.00	0.02 ^c^ ± 0.00 ^c^
p-coumaric acid	0.02 ^d^ ± 0.00	0.01 ± 0.00	0.02 ^d^ ± 0.00	- ^a,c,^*	0.01 ± 0.00
syringic acid	2.67 ^d^ ± 0.05	3.68 ^a,c,d,e^ ± 0.30	6.61 ^d,e^ ± 0.70	0.44 ^a,b,c,e^ ± 0.03	0.65 ^c,d^ ± 0.05
ferulic acid	0.114 ^b,c,d,e^ ± 0.00	0.03 ^a,c,d,e^ ± 0.00	0.319 ^a,b,d,e^ ± 0.03	- ^a,b,c,^*	- ^a,b,c,^*
salicylic acid	1.58 ^b,c,d,e^ ± 0.02	0.99 ^a,c,d,e^ ± 0.02	0.57 ^a,b,d,e^ ± 0.05	0.03 ^a,b,c,e^ ± 0.00	0.19 ^a,b,c,d^ ± 0.02
kaempferol	0.045 ^d^ ± 0.00	0.025 ^c^ ± 0.00	0.06 ^b,d,e^ ± 0.00	- ^a,c,e,^*	0.03 ^c,d^ ± 0.00

Statistically significant differences in means within groups (lines) are marked with letters. The test of differences among group means was conducted at *p* < 0.05 using the Scheffe’s post hoc procedure testing multiple comparisons available in ANOVA Statistica 13. * below detection limit; ±SD—standard deviation.

## Data Availability

Data are contained within the article.

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
