# Peer review of "Assessment of Phenolic Acid Content and Antioxidant Properties of the Pulp of Five Pumpkin Species Cultivated in Southeastern Poland"

_ijms, 2023, doi:10.3390/ijms24108621_

Round 1

Reviewer 1 Report

The manuscript entitle "Assessment of phenolic acid content and antioxidant properties of the pulp of five pumpkin species cultivated in south-eastern Poland" deals with the characterization of the phenolic content of 5 pumpkin cultivars and the evaluation of their antioxidant capacity. The manuscript is well written but in my opinion lacks of novelty. 

Lines 219-235. This is a detailed description of the soil and the cultivation method where the plant material growth. In my opinion it is not neccesary. It is written more focused to be an agronomic manuscript instead of a biofunctional manuscript.

Lines 239 and 241. The pumpkins were stored at -25ºC. It is common to used -20ºC and -80ºC. It is difficult to measured -25ºC. Can the author explained this?

Lines 243-248. Did the authors validated the extraction method? they used a big volumen (25mL) to extract 0.75g. How is it possible?

Lines 278-285. Why the authors carry out the total phenol content by the Folin-Ciocalteu method? This is an non precise methodology used when a HPLC-PDA-MS is not available. I suggested to remove the results and the mention to the Folin and also to the flavonoid content method, as both are spectrophotometrically methods and can not be compared with the HPLC-PDA.MS methodology. Also these method are not mentioned in the abstract.

Lines 299-310. The chromatographic method to determined the phenolic compounds in the pulp extracts should be validated in terms of specificity, linearity, LOQ, LOD, matrix effect and recovery.

Also, it is very strange that the phenolic compounds identified in the extracts are in their aglycone form and not bound to sugars, which is normal. Please, can the authors provide the HPLC-PDA-MS profiles and also the MS identification characteristics?

Author Response

Response to Reviewer 1 Comments

Manuscript ID:  ijms-2306374

Tytuł: Assessment of phenolic acid content and antioxidant properties of the pulp of five pumpkin species cultivated in south-eastern Poland

Dear Reviewer 1,

We are very grateful for your thorough review of our manuscript. The text has been corrected as per the remarks of Reviewer 1 and below please find comments to the remarks. We hope that the research results obtained from the study of the pulp of 5 pumpkin species will raise consumer awareness on the health benefits of the individual pumpkin species and will be a source of knowledge on selecting pumpkin species as a valuable raw material for food processing in order to develop new food products as an example of functional food. In addition, our research results constitute basic reference materials for further studies on the 5 pumpkin species cultivated in south-eastern Poland.

We hope that in its present form the manuscript meets the requirements of the International Journal of Molecular Sciences. We kindly ask Reviewer 1 to accept our replies.

Point 1. Lines 219-235. This is a detailed description of the soil and the cultivation method where the plant material growth. In my opinion it is not neccesary. It is written more focused to be an agronomic manuscript instead of a biofunctional manuscript

Response 1: Thank you for the remark

We removed the fragment on the soil and cultivation conditions from the manuscript.

Below is the text following the implementation of the remark of Reviewer 1:

3.1. Plant Material

The study material was the pulp of five pumpkin species: Cucurbita maxima "Bambino", Cucurbita pepo "Kamo Kamo", Cucurbita moschata "Butternut", Cucurbita ficifolia "Chilacayote Squash", Cucurbita argyrosperma "Chinese Alphabet". Plants were grown in Żyznów (49°49′ N 21°50′ E) (Poland) in 2018, under identical agro-climatic conditions. The pumpkin fruits were harvested on October 20 by cutting off the fruits from the shoot. Immediately after harvesting, the pumpkin fruits were washed, peeled and seeds were removed. The resulting pulp was crushed into small pieces, which were frozen at -25oC, and then freeze-dried (0.37 mBa) for 48 h in a FreeZone 12L freeze dryer (Labconco Corporation Kansas City, MO, USA). The obtained samples of lyophilisates were stored at -25°C in dark jars protected from light.

 Point 2. Lines 239 and 241. The pumpkins were stored at -25ºC. It is common to used -20ºC and -80ºC. It is difficult to measured -25ºC. Can the author explained this?

Response 2: Thank you for the remark

Dear Reviewer 1, immediately after harvesting, the pumpkin fruits were washed, peeled and seeds were removed. The resulting pulp was crushed into small pieces, which were frozen at -25C. When determining the temperature, we followed the information from the scientific paper published in the Molecules Journal:

Kostecka – Gugała, A.; Kruczek, M.; Ledwożyw-Smoleń, I.; Kaszycki, P. Antioxidants and Health-Beneficial Nutrients in Fruits of Eighteen Cucurbita Cultivars: Analysis of Diversity and Dietary Implications. Molecules 2020, 25, 1792; doi:10.3390/molecules25081792.

Our lab is equipped with a laboratory freezer, which can be set to -25°C (type: Arctiko LF 1400, -30oC).

Following the remark of Reviewer 1, we quoted the method of freezing of pumpkin samples: number [31]

Referencji number:

[31] Kostecka – Gugała, A.; Kruczek, M.; Ledwożyw-Smoleń, I.; Kaszycki, P. Antioxidants and Health-Beneficial Nutrients in Fruits of Eighteen Cucurbita Cultivars: Analysis of Diversity and Dietary Implications. Molecules 2020, 25, 1792; doi:10.3390/molecules25081792

Below please find the text with the authors' quote of the methodology of freezing pumpkin samples:

,,Immediately after harvesting, the pumpkin fruits were washed, peeled and seeds were removed. The resulting pulp was crushed into small pieces, which were frozen at -25oC, and then freeze-dried (0.37 mBa) for 48 h in a FreeZone 12L freeze dryer (Labconco Corporation Kansas City, MO, USA). The obtained samples of lyophilisates were stored at -25°C in dark jars protected from light [31].’’

We kindly ask that Reviewer 1 accepts our comment to remark point 2.

Point 3. Lines 243-248. Did the authors validated the extraction method? they used a big volumen (25mL) to extract 0.75g. How is it possible?

Response 3: Thank you for the remark

Dear Reviewer, we adopted the research methodology of the authors mentioned below. In our paper in section 3.2. Sample preparation, we quoted the methodology of those authors in point [31].

Kostecka – Gugała, A.; Kruczek, M.; Ledwożyw-Smoleń, I.; Kaszycki, P. Antioxidants and Health-Beneficial Nutrients in Fruits of Eighteen Cucurbita Cultivars: Analysis of Diversity and Dietary Implications. Molecules 2020, 25, 1792; doi:10.3390/molecules25081792

We kindly ask that Reviewer 1 accepts our response to the remark point 3.

Point 4. Lines 278-285. Why the authors carry out the total phenol content by the Folin-Ciocalteu method? This is an non precise methodology used when a HPLC-PDA-MS is not available. I suggested to remove the results and the mention to the Folin and also to the flavonoid content method, as both are spectrophotometrically methods and can not be compared with the HPLC-PDA.MS methodology. Also these method are not mentioned in the abstract.

Response 4: Thank you for the remark

Dear Reviewer 1, when writing the paper, we followed the principle of practicality. Therefore, we decided to measure the total phenol content by means of the Folin-Ciocalteu spectrophotometric method. The majority of production plants are only equipped with spectrophotometres rather than chromatographs. In this case, it is much easier and more feasible to measure the total phenol content by means of the aforementioned method, because the interested parties are able to quickly verify the biological activity based on the total phenol content. In the case of the chromatographic analysis, the  precision is higher, but the practical application  of the results will be low due to the unavailability of chromatographs in most companies in the fruit and vegetable processing industry.

According to the remark of Reviewer 1, we have included the missing elements in the abstract. Belowe please find the revised abstract.

,,Antioxidant properties and phenolic acid content in the pulp of five pumpkin species were evaluated. The following species were included: Cucurbita maxima 'Bambino', Cucurbita pepo 'Kamo Kamo', Cucurbita moschata 'Butternut', Cucurbita ficifolia 'Chilacayote Squash', Cucurbita argyrosperma 'Chinese Alphabet' cultivated in Poland. The content of polyphenolic compounds was determined using ultra-performance liquid chromatography coupled to photodiode array detection tandem mass spectrometry (UPLC-PDA-MS), the content of total phenols and total flavonoids was determined by spectrophotometric methods and the antioxidant activity was assessed by DPPH and FRAP assays. Ten phenolic compounds (protocatechuic acid, p-hydroxybenzoic acid, catechin, chlorogenic acid, caffeic acid, p-coumaric acid, syringic acid, ferulic acid, salicylic acid, kaempferol) were identified. Phenolic acids were the most abundant compounds; the amount of syringic acid was found to be the highest ranging from 0.44 (C. ficifolia) to 6.61 mg∙100 g-1 FW (C. moschata). Moreover, two flavonoids, catechin and kaempferol, were detected; their highest content was found in the C. moschata pulp (catechins: 0.31 mg∙100 g-1 FW and kaempferol: 0.06 mg ∙100 g-1 FW); the lowest amounts were detected in C. ficifolia (catechins- 0.15 mg ∙100 g-1 FW and kaempferol below the limit of detection). Analysis of the antioxidant potential showed significant differences depending on the species and the test used. The DPPH radical scavenging activity of C. maxima was 1.03 times higher, as compared to the C. ficiofilia pulp, and 11. 60 times higher, as compared to C. pepo. In the case of FRAP assay, the multiplicity of FRAP radical activity in the C. maxima pulp was 4.65 times higher, as compared to the C. Pepo pulp and only 1.08 times higher, as compared to the C. ficifolia pulp. The study findings show a high health-promoting value of the pumpkin pulp; however, the content of phenolic acids and antioxidant properties are species-dependent’’.

Point 5. Lines 299-310. The chromatographic method to determined the phenolic compounds in the pulp extracts should be validated in terms of specificity, linearity, LOQ, LOD, matrix effect and recovery.

Response 5: Thank you for the remark

Dear Reviewer 1, due to the use of routine phenolic compound analyses by the device (chromatograph HPLC Shimadzu LC-10AS), validation took place in November last year and involved the validation of the obtained results basen on protocol Q2(R1) Validation of Analytical Procedured Text and Methodology. Guidance for Industry.

Point 6. Also, it is very strange that the phenolic compounds identified in the extracts are in their aglycone form and not bound to sugars, which is normal. Please, can the authors provide the HPLC-PDA-MS profiles and also the MS identification characteristics?

Response 6: Thank you for the remark

Dear Reviewer 1, the analyses were Carrier out by means of chromatograph HPLC Shimadzu LC-10AS with photodiode detector SPD-10AV UV-VIS. Phenolic  compounds were identifued by means of referencji materials and detection times. We were unable to run analyses by means of HPLC-PDA-MS.

Yours Sincerely,

Dr hab. inż. Barbara Krochmal-Marczak

Department of Plant Production and Food Safety

The University College of Applied Sciences in Krosno

38-400 Krosno, Poland

email: [email protected]

dr inż. Tomasz Cebulak

Department of Food Technology and Human Nutrition,

Institute of Food Technology and Nutrition,

College of Natural Sciences,

University of Rzeszów, 35-601 Rzeszów, Poland

Reviewer 2 Report

Specific points:

In my opinion, nothing new was given in this research that would be scientifically significant. Namely, I believe that in addition to the composition of biologically active components in fresh pumpkin pulp (which depends on the type of pumpkin, which is actually expected), the content and activity of these components largely depends on the applied processing system during food preparation. Therefore, if we look at the nutritional value of pumpkin pulp, it should be examined through the prism of the specific treatment applied. Because no one consumes pumpkin in its fresh state (unprocessed). So the data in this research would only be a starting point for further research, where knowledge would be gained about the nutritional value of the product being consumed.

Otherwise, I think that the authors have presented the obtained results in a reviewed and satisfactory way, but explanations are missing.

 L: 270 – “…according to the method described by Zhou et al. [23]…” - In the list of references, this reference is listed as number 22. The statements of all other references should be checked.

L: “Our findings should increase the consumers awareness of the health-promoting qualities of individual pumpkin species and provide knowledge on the selection of pumpkin species as a valuable raw material for food processing in order to develop new functional food products for adults and baby food.” - This statement is not entirely true. The authors can be sure of this claim if it has been observed how the content/activity of the components changes after the applied process of production of functional food or baby food. As such, pumpkin pulp represents only a potentially good raw material for the production of these foods.

As I mentioned above, I think that a certain way of processing pumpkin pulp (the samples mentioned here) should be done, and then compare and explain what results/changes occur. Otherwise, in my opinion, these results (of which components there are more or less) are not sufficient by themselves to represent scientific work.

Accordingly, the Introduction should also provide an overview of how the activity and content of the biologically active components of the pumpkin pulp changes depending on the applied treatment.

_________________

All my suggestions are for improving the manuscript. I hope all the suggestions are clear.

Best regards.

Author Response

Response to Reviewer 2 Comments

Manuscript ID:  ijms-2306374

:

Tytuł: Assessment of phenolic acid content and antioxidant properties of the pulp of five pumpkin species cultivated in south-eastern Poland

Dear Reviewer 2,

We are very grateful for your thorough review of our manuscript. The remarks of Reviewer 2 have been implemented into the text and have significantly increased its scientific value. The text has been corrected as per the reviewers’ remarks, and below please find replies to the remarks of Reviewer 2. We hope that in its present form the manuscript meets the requirements of the International Journal of Molecular Sciences.

Point 1. In my opinion, nothing new was given in this research that would be scientifically significant. Namely, I believe that in addition to the composition of biologically active components in fresh pumpkin pulp (which depends on the type of pumpkin, which is actually expected), the content and activity of these components largely depends on the applied processing system during food preparation. Therefore, if we look at the nutritional value of pumpkin pulp, it should be examined through the prism of the specific treatment applied. Because no one consumes pumpkin in its fresh state (unprocessed). So the data in this research would only be a starting point for further research, where knowledge would be gained about the nutritional value of the product being consumed.

Response 1: Thank you for the remark

We completely agree with this valuable remark of the esteemed Reviewer.  The research results that we present are merely the first step in a larger series of studies on various methods of processing the discussed pumpkin species. Furthermore, we intend future publications to be based on our results, so this paper shall be quoted. However, we believe that a detailed analysis of the health benefits of several pumpkin species is highly relevant, in particular in the context of obtaining raw material from identical cultivation conditions. The awareness of the health properties of pumpkin pulp can increase consumer interest in the consumption of this valuable vegetable and can bring tangible benefits for farmers who will become interested in pumpkin cultivation.

 Point 2. L: 270 – “…according to the method described by Zhou et al. [23]…” - In the list of references, this reference is listed as number 22. The statements of all other references should be checked.

Response 2: Thank you for the remark

We would like to thank you for noticing our mistake in the numbering of the quoted references. We have corrected that and the right number has been inserted into the main text. Numbers have been checked in the entire manuscript.

Point 3. L: “Our findings should increase the consumers awareness of the health-promoting  potential of individual pumpkin species and provide knowledge on the selection of pumpkin species as a valuable raw material for food processing in order to develop new functional food products for adults and baby food.” - This statement is not entirely true. The authors can be sure of this claim if it has been observed how the content/activity of the components changes after the applied process of production of functional food or baby food. As such, pumpkin pulp represents only a potentially good raw material for the production of these foods.

Response 3: Thank you for the remark

Indeed, this sentence was overstated. That is why we have added that it has health-promoting potential. Thank you for this valuable remark. We have corrected it and it is no longer misleading to the reader.

Point 4. As I mentioned above, I think that a certain way of processing pumpkin pulp (the samples mentioned here) should be done, and then compare and explain what results/changes occur. Otherwise, in my opinion, these results (of which components there are more or less) are not sufficient by themselves to represent scientific work.

Response 4: Thank you for the remark

 As we have already mentioned before, this is just the first step in a series of research, which are in progress, and we intend to publish them in separate articles due to the large volume of the results. The aim of the reviewed paper is to conduct research on fresh raw material. The aim of the results is to raise the awareness of consumers and the food industry of the health benefits of particular pumpkin species. The authors agree that the identification of the basic potential of pumpkin pulp is sufficient for publication, which shall be the basis for future articles on pumpkin pulp processing.

Point 5. Accordingly, the Introduction should also provide an overview of how the activity and content of the biologically active components of the pumpkin pulp changes depending on the applied treatment.

Response 5: Thank you for the remark

We would like to thank Reviewer 2 for the valuable remark on including in the Introduction an overview of how the activity and content of the biologically active components of the pumpkin pulp changes depending on the applied thermal treatment. This remark shall be investigated further in future research studies. The current studies and results are our first research in a series of research on the health benefits of five pumpkin species: Cucurbita maxima “Bambino”, Cucurbita pepo “Kamo Kamo”, Cucurbita moschata “Butternut”, Cucurbita ficifolia “Chilacayote Squash”, Cucurbita argyrosperma “Chinese Alphabet”. The research was based on fresh raw material. We intend to conduct more research on the influence of various thermal treatment methods (cooking, blanching) on selected physical properties and health benefits of five pumpkin species: Cucurbita maxima “Bambino”, Cucurbita pepo “Kamo Kamo”, Cucurbita moschata “Butternut”, Cucurbita ficifolia “Chilacayote Squash”, Cucurbita argyrosperma “Chinese Alphabet”. If the results of the current study of fresh material are published in the International Journal of Molecular Sciences, they will be used are reference for comparison with other research results, which we intend to conduct on the subject matter suggested by the Reviewer, i.e. the influence of thermal treatment (cooking, blanching) on the bioactive compounds in the pulp of five pumpkin species.

Yours Sincerely,

Dr hab. inż. Barbara Krochmal-Marczak

Department of Plant Production and Food Safety

The University College of Applied Sciences in Krosno

38-400 Krosno, Poland

email: [email protected]

dr inż. Tomasz Cebulak

Department of Food Technology and Human Nutrition,

Institute of Food Technology and Nutrition,

College of Natural Sciences,

University of Rzeszów, 35-601 Rzeszów, Poland

Round 2

Reviewer 1 Report

The authors have partially answered my comments. 

I´m not agree with the comments regarding the Folin and total flavonoid content. This is a scientific manuscript and not a comunication for plant productors. 

Moreover, the authors have said that they were unable to run the analysis using a UPLC-PDA-MS, so Why they include this in the abstract (line 20)?

Also, if they run the validation method in november, they should include the values of this validation (specificity, linearioty, LOQ, LOD, matrix effect and recovery). Moreover, it is neccesary to show a figure with the HPLC-PDA chromatograms of the identified compound in the plants. 

Author Response

Dear Editor!

Dear Reviewer 1!

Thank you for your evaluation of our manuscript “Assessment of phenolic acid content and antioxidant properties of the pulp of five pumpkin species cultivated in south-eastern Poland”.

               We are grateful for your valuable remarks and suggestions during the first and second review. We are certain that the remarks and suggestions of Reviewer 1 and Reviewer 2 will significantly improve the quality of our manuscript. Following the first review, the article has been changed according to the suggestions and remarks of Reviewer 1 and Reviewer 2; we have made all the improvements and modifications suggested by the Reviewers, and the changes in the text are marked in yellow:

  • we have included the Reviewer’s remark in the abstract;
  • we have corrected the numbering of references in the text;
  • we have checked reference numbers in the entire manuscript;
  • we have removed the description of the weather and soil conditions;
  • we have explained our methodology of pumpkin storage at

-25C and quoted scientific sources referring to this methodology;

  • according to the remark, in the result summary, we have added that it has “health-promoting potential”. Thank you for this valuable remark. We have corrected it and it is no longer misleading to the reader.
  • In the reference list, we have translated item no. 30 into English.
  • We have also replied to the Reviewers’ suggestions and comments.

Dear Editor, we would also like to thank you for the second review.

Below please find our replies to the remarks of Reviewer 1 and Reviewer 2.

Replies to the remarks and comments of Reviewer 1

Point 1: “,I´m not agree with the comments regarding the Folin and total flavonoid content. This is a scientific manuscript and not a comunication for plant productors.”

Reply to remark 1: Upon a thorough analysis of research methods in scientific articles published in scientific journals, it turns out that the Folin-Ciocalteu method is a reference method recognised in a number of research studies. The International Journal Molecules Science has also published a great deal of scientific papers using the Folin–Ciocalteu method. Below please find a selection of scientific articles published in the International Journal Molecules Science that include the Folin–Ciocalteu method in their research methodologies.

Int. J. Mol. Sci. 2023, 24(4), 3558; https://doi.org/10.3390/ijms24043558

Int. J. Mol. Sci. 2023, 24(1), 54; https://doi.org/10.3390/ijms24010054

Int. J. Mol. Sci. 2022, 23(22), 13907; https://doi.org/10.3390/ijms232213907

Int. J. Mol. Sci. 2019, 20(22), 5560; https://doi.org/10.3390/ijms20225560

Int. J. Mol. Sci. 2018, 19(8), 2255; https://doi.org/10.3390/ijms19082255

Int. J. Mol. Sci. 2017, 18(12), 2434; https://doi.org/10.3390/ijms18122434

  1. J. Mol. Sci. 2017, 18(2), 376; https://doi.org/10.3390/ijms18020376

We kindly ask Reviewer 1 to accept this method in our research.

Point 2: Moreover, the authors have said that they were unable to run the analysis using a UPLC-PDA-MS, so Why they include this in the abstract (line 20)?

Reply to remark 2: Thank you for the remark of Reviewer 2.

We have corrected the abstract and added the following:

Correct text:

“The content of β-carotene, vitamin E, and phenolic compounds was measured with the High-Performance Liquid Chromatography (HPLC), whereas the total polyphenols, total flavonoids, and antioxidant properties were determined by means of spectrophotometric methods.

 We kindly ask Reviewer 1 to accept our reply.

Point 3. Also, if they run the validation method in november, they should include the values of this validation (specificity, linearioty, LOQ, LOD, matrix effect and recovery). Moreover, it is neccesary to show a figure with the HPLC-PDA chromatograms of the identified compound in the plants. 

Reply to remark 3: Dear Reviewer, thank you for the remark, we deem it right. Due to very vast and versatile research subjects, so far we have not archived our research studies and we have no matrix. However, you are right and from now on, we will follow the remark of Reviewer 1.  We hope that lack of such documentation will not affect the quality of our results, as we have provided the research methodology, references, and equipment characteristics. We have also shown statistical differences or lack thereof in analyses performed on the same day.

We kindly ask the Editor, Reviewer 1 to accept our manuscript for publication in the International Journal Molecules Science, Section “Bioactives and Nutraceuticals”, Special Issue “Role of Polyphenols in Human Health and Food Systems”.

Reviewer 2 Report

Specific points:

I would like to thank the authors for their responses to my comments. However, no significant changes have been made to the work itself. I simply believe that only the analysis of the composition of the raw material (in the sense that there is more or less of something) cannot be sufficient for scientific work. In my opinion, it is only the basis for conducting research that would have a new and scientific dimension. This especially applies to raw materials that are not consumed as such in their raw state.

That's why I still stand by my position that all these analytical studies should be done on samples that would be processed and available to consumers as such.

__________________

All my suggestions are for improving the manuscript. I hope all the suggestions are clear.

Best regards.

Author Response

Dear Editor!

 Dear Reviewer 2!

Thank you for your evaluation of our manuscript “Assessment of phenolic acid content and antioxidant properties of the pulp of five pumpkin species cultivated in south-eastern Poland”.

               We are grateful for your valuable remarks and suggestions during the first and second review. We are certain that the remarks and suggestions of Reviewer 1 and Reviewer 2 will significantly improve the quality of our manuscript. Following the first review, the article has been changed according to the suggestions and remarks of Reviewer 1 and Reviewer 2; we have made all the improvements and modifications suggested by the Reviewers, and the changes in the text are marked in yellow:

  • we have included the Reviewer’s remark in the abstract;
  • we have corrected the numbering of references in the text;
  • we have checked reference numbers in the entire manuscript;
  • we have removed the description of the weather and soil conditions;
  • we have explained our methodology of pumpkin storage at

-25C and quoted scientific sources referring to this methodology;

  • according to the remark, in the result summary, we have added that it has “health-promoting potential”. Thank you for this valuable remark. We have corrected it and it is no longer misleading to the reader.
  • In the reference list, we have translated item no. 30 into English.
  • We have also replied to the Reviewers’ suggestions and comments.

Dear Editor, we would also like to thank you for the second review.

Below please find our replies to the remarks of Reviewer 1 and Reviewer 2.

Replies to the remarks and comments of Reviewer 1

Point 1: “,I´m not agree with the comments regarding the Folin and total flavonoid content. This is a scientific manuscript and not a comunication for plant productors.”

Reply to remark 1: Upon a thorough analysis of research methods in scientific articles published in scientific journals, it turns out that the Folin-Ciocalteu method is a reference method recognised in a number of research studies. The International Journal Molecules Science has also published a great deal of scientific papers using the Folin–Ciocalteu method. Below please find a selection of scientific articles published in the International Journal Molecules Science that include the Folin–Ciocalteu method in their research methodologies.

Int. J. Mol. Sci. 2023, 24(4), 3558; https://doi.org/10.3390/ijms24043558

Int. J. Mol. Sci. 2023, 24(1), 54; https://doi.org/10.3390/ijms24010054

Int. J. Mol. Sci. 2022, 23(22), 13907; https://doi.org/10.3390/ijms232213907

Int. J. Mol. Sci. 2019, 20(22), 5560; https://doi.org/10.3390/ijms20225560

Int. J. Mol. Sci. 2018, 19(8), 2255; https://doi.org/10.3390/ijms19082255

Int. J. Mol. Sci. 2017, 18(12), 2434; https://doi.org/10.3390/ijms18122434

  1. J. Mol. Sci. 2017, 18(2), 376; https://doi.org/10.3390/ijms18020376

We kindly ask Reviewer 1 to accept this method in our research.

Point 2: Moreover, the authors have said that they were unable to run the analysis using a UPLC-PDA-MS, so Why they include this in the abstract (line 20)?

Reply to remark 2: Thank you for the remark of Reviewer 2.

We have corrected the abstract and added the following:

Correct text:

“The content of β-carotene, vitamin E, and phenolic compounds was measured with the High-Performance Liquid Chromatography (HPLC), whereas the total polyphenols, total flavonoids, and antioxidant properties were determined by means of spectrophotometric methods.

 We kindly ask Reviewer 1 to accept our reply.

Point 3. Also, if they run the validation method in november, they should include the values of this validation (specificity, linearioty, LOQ, LOD, matrix effect and recovery). Moreover, it is neccesary to show a figure with the HPLC-PDA chromatograms of the identified compound in the plants. 

Reply to remark 3: Dear Reviewer, thank you for the remark, we deem it right. Due to very vast and versatile research subjects, so far we have not archived our research studies and we have no matrix. However, you are right and from now on, we will follow the remark of Reviewer 1.  We hope that lack of such documentation will not affect the quality of our results, as we have provided the research methodology, references, and equipment characteristics. We have also shown statistical differences or lack thereof in analyses performed on the same day.

Replies to the remarks and comments of Reviewer 2

Point 1. “I would like to thank the authors for their responses to my comments. However, no significant changes have been made to the work itself. I simply believe that only the analysis of the composition of the raw material (in the sense that there is more or less of something) cannot be sufficient for scientific work. In my opinion, it is only the basis for conducting research that would have a new and scientific dimension. This especially applies to raw materials that are not consumed as such in their raw state. That's why I still stand by my position that all these analytical studies should be done on samples.”

Reply to remark 1: Dear Reviewer 2,

               The aim of our paper was to identify and evaluate the content of phenolic acids and antioxidant properties in the pulp of five pumpkin species: C. “Bambino”, C. pepo “Kamo Kamo", C. moschata “Butternut", C. ficifolia “Chilacayote Squash”, C. argyrosperma “Chinese Alphabet" cultivated in south-eastern Poland. The results of our research have shown that the pulp of the selected pumpkin species is the source of phenolic acids and demonstrates antioxidant properties; however, the total content of the identified compounds in the pumpkin species is different. For the consumer, the results of our research are a valuable source of information on potential health properties of pumpkin and will allow for making informed decisions on the consumption of pumpkin species with the greatest health properties. Furthermore, our research results have practical application, as they provide knowledge on the health properties of pumpkin as a valuable raw material for food processing in order to develop new functional foods. Once published, our research results will be a source of useful information for scientists focusing on the health potential of pumpkin pulp. The research results that we have presented are applicable, which is of high importance in terms of science. Unfortunately, in this paper, we are unable to present new research studies on processed raw material, because our methodology and research were based on fresh raw material (which is suitable for consumption as raw food, just like raw material for thermal processing). Besides, the research suggested by Reviewer 2 requires a completely different sample preparation methodology. Moreover, new research results require a new statistical analysis and table preparation, as well as a new description of results. We believe it would be too much data in a single paper and we intend to publish such data in a separate article. 

               We are unable to deliver such a paper within 10 days. However, we would like to thank Reviewer 2 for the suggestion. As we have already mentioned in the replies to the first review, we intend to conduct further research on the influence of various thermal processing techniques (cooking, blanching) on the selected physical properties and health potential of the pulp of the pumpkin species under study.

We kindly ask Reviewer 2 to accept our replies.

We kindly ask the Editor, Reviewer 2 to accept our manuscript for publication in the International Journal Molecules Science, Section “Bioactives and Nutraceuticals”, Special Issue “Role of Polyphenols in Human Health and Food Systems”.

Round 3

Reviewer 2 Report

Specific points:

I would like to thank the authors for their responses to my comments. However, no significant changes have been made to the work itself. I simply believe that only the analysis of the composition of the raw material (in the sense that there is more or less of something) cannot be sufficient for scientific work. In my opinion, it is only the basis for conducting research that would have a new and scientific dimension. This especially applies to raw materials that are not consumed as such in their raw state.

That's why I still stand by my position that all these analytical studies should be done on samples that would be processed and available to consumers as such.

 __________________

 All my suggestions are for improving the manuscript. I hope all the suggestions are clear.

 Best regards.

Author Response

Dear Editor, 

Dear Reviewer 2,

Thank you very much for your kind review 3. In the review, the reviewer suggests that ours should be about heat-treated raw material, which was not our research objective. Indeed, in the submitted manuscript, this objective was very narrowly presented, hence perhaps Reviewer 2's suggestion. Therefore, in the current version, we have added information about the objectives of our research to the introduction. The content of the objective of our research is as follows:

"Considering the fact that more and more new varieties and species of pumpkins differing in shape, color and flavor are being grown and sold, we undertook a study to gain knowledge of the antioxidant properties and levels of phenolic acids in the fresh flesh of five pumpkin species grown in southeastern Poland. An additional goal of our research is to provide knowledge of raw pumpkin flesh as a valuable raw material for the pharmaceutical, cosmetic and food industries''.

  • In addition, we expanded the introduction of the mauscript to include studies by other authors that report on the high health-promoting qualities of raw pumpkin flesh and the potential for use in the pharmaceutical, cosmetic and food industries.
  • We have supplemented in the literature list the references to which we expanded the introduction.
  • We have corrected the numbering of the cited literature in our manuscript.
  • We have supplemented the conclusions with the results of our stated research objectives.

We have highlighted all changes in yellow.

Dear reviewer 2, we believe that the results of our study of fresh pumpkin flesh are a valuable source of information on the different levels of antioxidant properties and phenolic acids of the flesh of the studied pumpkin species grown under the soil and climatic conditions of Poland. In addition, the results of our research are of an applied nature, as they provide the pharmaceutical and cosmetic industries with knowledge of the health-promoting properties of pumpkin flesh as a valuable raw material for the production of extracts for the development of new medicinal products, as well as a valuable source of information for food processing to develop new food products as an example of functional foods in both adult and child nutrition. The research results we have presented will increase consumer awareness of the health-promoting properties of various pumpkin species, and can also be a source of knowledge regarding the selection for cultivation of pumpkin species with the highest antioxidant properties and the highest content of phenolic acids. The published results of our study can also contribute to further research on these pumpkin species for health benefits.

Dear Reviewer 2,

Please accept the papers from our research, as we are experts in this field. Our works are published in journals of high scientific value. The scientific output of all of our entire scientific team is related to laboratory quality assessment studies of fresh plant raw materials, not processed raw materials. We will keep in mind the remark of Reviewer 2, but this requires supplementing our team with experts and scientists from the field of scientific food processing. Besides, the research that Reviewer 2 suggests to us requires a completely different sample preparation methodology. 

At present, we do not have the research material at our disposal, as it is a one-year raw material, and harvesting begins in autumn. In addition, the new research results require statistical and tabular reworking, as well as a new description of the results. We are not able to submit such a study in 10 days.

However, we thank Reviewer 2 for this suggestion, and as we responded in the amendments after the first and second reviews, we plan with an expanded research team after the harvest of the raw material to conduct further studies on the effects of different heat treatment techniques (cooking, blanching) on selected physical characteristics and the health-promoting value of the flesh of the pumpkin species studied.

Below are some of our published works:

Krochmal-Marczak, B.; Cebulak, T.; Kapusta, I.; Oszmiański, J.; Kaszuba, J.; Żurek, N. The Content of Phenolic Acids and Flavonols in the Leaves of Nine Varieties of Sweet Potatoes (Ipomoea batatas L.) Depending on Their Development, Grown in Central Europe. Molecules 2020, 25, 3473. doi: 10.3390/molecules25153473

Krochmal-Marczak, B.; Sawicka, B.; Krzysztofik, B.; Danilčenko, H.; Jariene, E. The Effects of Temperature on the Quality and Storage Stalibity of Sweet Potato (Ipomoea batatas L. [Lam]) Grown in Central Europe. Agronomy 2020, 10, 1665. doi: 10.3390/agronomy10111665

Sawicka, B.; Krochmal-Marczak, B.; Pszczółkowski, P.; Bielińska, E.J.; Wójcikowska-Kapusta, A.; Barbaś, P.; Skiba, D. Effect of Differentiated Nitrogen Fertilization on the Enzymatic Activity of the Soil for Sweet Potato (Ipomoea batatas L. [Lam.]) Cultivation. Agronomy 2020, 10, 1970. doi: 10.3390/agronomy10121970

Sawicka, B.; Pszczółkowski, P.; Kiełtyka-Dadasiewicz, A.; Barbaś, P.; Ćwintal, M.; Krochmal-Marczak, B. The Effect of Effective Microorganisms on the Quality of Potato Chips and French Fries. Appl. Sci. 2021, 11, 1415. doi: 10.3390/app11041415

Sawicka, B.; Śpiewak, M.; Kiełtyka-Dadasiewicz, A.; Skiba, D.; Bienia, B.; Krochmal-Marczak, B.; Pszczółkowski, P. Assessment of the Suitability of Aromatic and High-Bitter Hop Varieties (Humulus lupulus L.) for Beer Production in the Conditions of the Małopolska Vistula Gorge Region. Fermentation 2021, 7, 104. doi: 10.3390/fermentation7030104

Pszczółkowski, P.; Krochmal-Marczak, B.; Sawicka, B.; Pszczółkowski, M. The Impact of Effective Microorganisms on Flesh Color and Chemical Composition of Raw Potato Tubers. Appl. Sci. 2021, 11, 8959. doi: 10.3390/app11198959

Sawicka, B.; Barbaś, P.; Pszczółkowski, P.; Skiba, D.; Yeganehpoor, F.; Krochmal-Marczak, B. Climate Changes in Southeastern Poland and Food Security. Climate 2022, 10, 57. doi: 10.3390/cli10040057

Cebulak, T.; Krochmal-Marczak, B.; Stryjecka, M.; Krzysztofik, B.; Sawicka, B.; Danilčenko, H.; Jarienè, E. Phenolic Acid Content and Antioxidant Properties of Edible Potato (Solanum tuberosum L.) with Various Tuber Flesh Colours. Foods 2023, 12, 100. doi: 10.3390/foods12010100

Skiba, D.; Jariene, E.; Barbaś, P.; Krochmal-Marczak, B.; Sawicka, B. The Effect of Fertilization on the Structure of the Aboveground Biomass of Several Cultivars of Jerusalem Artichoke (Helianthus tuberosus L.). Agronomy 2023, 13, 314. doi: 10.3390/agronomy13020314

We kindly ask Reviewer 2 to accept our replies.

We kindly ask the Editor, Reviewer 2 to accept our manuscript for publication in the International Journal Molecules Science, Section “Bioactives and Nutraceuticals”, Special Issue “Role of Polyphenols in Human Health and Food Systems
